

# Technical Note: Incorporating topographic deflection effects into thermal history modelling

Richard A. Ketcham[1]

[1]Jackson School of Geosciences, University of Texas at Austin, Austin, TX 78712, USA

*Correspondence to*: Richard A. Ketcham (ketcham@jsg.utexas.edu)

**Abstract.** This contribution describes a set of equations and relations to calculate accurate cooling paths through the 2D temperature field of an exhuming region with periodic topography. A 1D model adequately captures the time-varying component of the system regardless of the complexity of the cooling path, making the computation efficient. A series of 2D finite element models demonstrate how temperatures below the periodic mid-slope, or mean topography, can be mapped to those below ridges and valleys, and how these transitions vary with topographic period and amplitude and the ratio of the near-surface geotherm to the atmospheric lapse rate. These new calculations are implemented into HeFTy to support multi-sample modelling of samples collected along topographic profiles, particularly for terranes with long-lived topography that exhumed through an inflected temperature field.

## 1 Introduction

High degrees of exhumation are characteristically accompanied by development of variable topography. During mountain building different areas of the crust rise or fall unevenly due to fault movements and block rotations. Erosion counteracts these incompletely, as erosion rates are a function of relief, and topography approaches a steady state condition while deformation continues (Adams, 1980; Willett and Brandon, 2002). Undulating topography in turn deflects isotherms below, increasing the space between geotherms below local highs and compressing them below local lows (Figure 1).

It is thus natural that the effects of topography on cooling rates, and how they can be used to infer exhumation rates, has long been of interest to thermochronologists. Early treatments of the effect of sinusoidal topography on steady-state thermal structure were provided by Birch (1950), Carslaw and Jaeger (1959), and Turcotte and Schubert (1982). Stüwe et al. (1994) expanded this work to provide equations for steady-state isotherms given a constant erosion rate. Mancktelow and Grasemann (1997) further broadened the set of available analytical solutions to approximate the time-dependent development of isotherms under constant sinusoidal topography after the onset of erosion at a constant rate. To study more general cases Mancktelow and Grasemann (1997) utilized a 2D finite difference model, and most subsequent work has gone into the development of 2D and 3D numerical solutions (e.g., Almendral et al., 2015; Braun, 2003), although further development of analytical solutions has continued as well (Fox et al., 2014; Fox et al., 2013).



Thermal history modelling consists of posing a large number of variable time-temperature (*t-T*) paths, calculating the response
of thermochronometric systems, and comparing the results to measured data to investigate which histories the data are
compatible with (Gallagher, 2012; Ketcham, 2005, 2024). The histories posed are of arbitrary complexity, with cooling rates
that vary rather than remain constant.

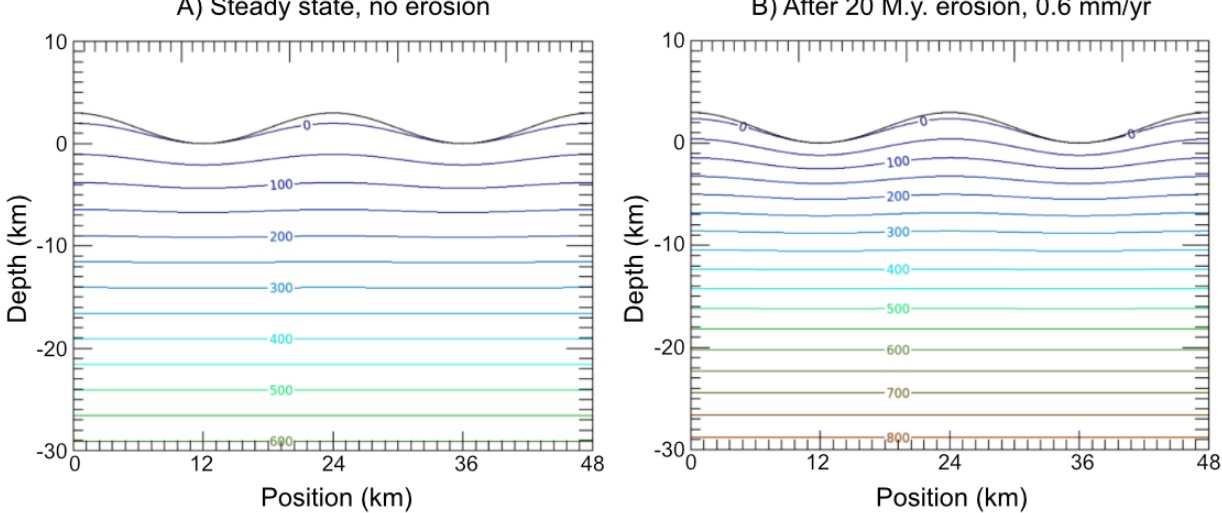

**Figure 1. Isotherms for periodic topography with wavelength 24 km, amplitude 1500 m, and steady-state geotherm of
20°C/km, and atmospheric lapse rate of 5 °C/km. A) Steady-state condition. B) After 20 million years of exhumation at
0.6 mm/yr. Calculated using FETKin (Almendral et al., 2015).**

A recent update to the HeFTy thermal history modelling software provided for the capability to perform inversions in time-
depth space (*t-z*), rather than time-temperature, by utilizing a 1D finite difference model to convert *t-z* paths to *t-T*, under the
assumption that changes in depth correspond to erosion or deposition at the surface (Ketcham, 2024). It was specifically
intended to help with simultaneous modelling of samples along topographic profiles. It allows for topographic development,
and relative vertical rotation (tilting) of sample positions with respect to each other during exhumation. The 1D model is
computationally fast enough that models can be calculated interactively, though the graphic interface.

A shortcoming of this implementation is that it neglects the effects of topography on thermal structure, thus treating exhumation
through ridges and valleys as equivalent. These effects could be roughly approximated by including topographic development,
starting a model as a peneplain and then increasing topographic amplitude during exhumation (e.g., Hestnes et al., 2024;
Mackaman-Lofland et al., 2024). However, this shortcut is non-ideal for studying areas with long term or steady state
topography.

This contribution presents a method to approximate the deflection effects of periodic topography for complex exhumation and
topographic development histories, using only the 1D thermal model calculation. It builds upon, and is informed by, previous
work described by Turcotte and Schubert (1982) and Mancktelow and Grasemann (1997). It is partly empirical, utilizing a





series of 2D finite element models to map out how variations in various parameters (wavelength, amplitude, geotherm, exhumation rate, topographic development) affect thermal structure, to refine analytical approximations and develop a set of

relations encompassing the range of conditions likely to be encountered in HeFTy-type thermal history modelling.

## 2 Methods

Thermal calculations for this work are performed using the FETKin 2D finite element model (Almendral et al., 2015). Grid spacing was 1 km, except for models with topographic wavelengths of 8 km, for which accuracy was improved by reducing to 0.5 km. All models used a constant basal flux boundary condition, began at steady state, and then run for 12 km of

exhumation, all with thermal conductivity $K$=2.5 W m$^{-1}$ K$^{-1}$, thermal diffusivity $\kappa$=9.26e-7 m$^2$ s$^{-1}$, and atmospheric lapse rate $\beta$=0.005 °C m$^{-1}$.

## 3 Model Development and Description

### 3.1 Steady state geotherm with constant periodic topography

For steady-state thermal conditions beneath periodic topography, Turcotte and Schubert (1982, Eq. 4-66, excluding heat

production) provide:

$$T(z) = T_0 + \frac{q_m z}{K} + \left(\frac{q_m}{K} - \beta\right) H_0 \cos\left(\frac{2\pi x}{\lambda}\right) e^{-2\pi z/\lambda} \tag{1}$$

where $z$ is depth with mean surface = 0 and positive upwards[1], $T_0$ is surface temperature at mean elevation, $q_m$ is mantle heat flow (W/m$^2$), $H_0$ is the amplitude of topography above and below the mean (m), $\lambda$ is topographic wavelength (m), and $x$ is position along the surface (m). Their solution is based on assuming the separability of the different components of the heat flow equation: the first term is the mean surface temperature, the second represents heat flow from depth (the steady-state

geotherm $g = q_m/K$), and the third reflects the transition in thermal gradient from the atmospheric lapse rate at the surface to the steady-state geotherm as a function of wavelength, dropping by factor of 1/e for every $\lambda/2\pi$ increase in depth.

This equation is built upon a simpler one that posits a flat Earth surface with periodically varying temperature. To incorporate topography, they make the simplifying assumption that topography is shallow (Turcotte and Schubert, 1982, p. 153), and use only the first term of the Taylor series expansion to extrapolate from the mean surface. Although adequate for illustrating

crust-scale patterns, this assumption does not reproduce the surface temperature at and immediately below the crests and valleys of periodic topography, because the derivation assumes the geotherm is linear at $z$=0 while at the same time has the topographic effect exponentially diminish with depth (and amplify with elevation) away from $z$=0.

---

[1] The ordering in the parenthetical term preceding $H_0$ is opposite in Mancktelow and Grasemann (1997) compared to Turcotte and Schubert (1982), making $z$ positive upwards, whereas in the original model setup $z$ is positive downwards. This work uses the Mancktelow and Grasemann (1997) ordering and convention.



The solution is improved by introducing a new variable $z_x$ representing depth below the local surface:

$$T(z_x) = T_0 + \frac{q_m z}{K} + \left(\frac{q_m}{K} - \beta\right) H_0 \cos\left(\frac{2\pi x}{\lambda}\right) e^{-2\pi z_x/\lambda} \tag{2}$$

where

$$z_x = z + H_0 \cos\left(\frac{2\pi x}{\lambda}\right) \tag{3}$$

Or in other words, for the purpose of topographic deflection, the Taylor series originates at the local surface rather than at mean topography.

Another consequence of utilizing separability is that it assumes that the geothermal gradient at the topographic midpoint ($z$=0) is the same as the geothermal gradient without any topography at all. Finite element modelling shows that this is not quite the case, as, in effect, valleys cool the crust more than hills allow it to retain heat. This effect is usually small, but grows with the

severity of topography (increasing $H_0$ and/or decreasing $\lambda$).

These two effects can be seen by comparing Equation (2) to a finite element model. Figure 2A shows the difference in the steady state geotherm between model and analytical solution below the crest, mid-slope, and valley of topography with wavelength 24 km and amplitude 1500 m. The $z_x$ shift in variable allows the temperatures to closely replicate in the near-surface, but the periodic topography cools the lower crust by ~4°C in the FE model compared to the analytical solution. All

1D solutions for areas of varying topography share this bias, but a simple correction can remove most of it.

Fig. 2B shows the offset of the mid-slope geotherm for a number of additional scenarios varying in topographic wavelength and amplitude, and geothermal gradient. All are fit well by the relation $T_{off}(z) = T_{off,\infty}(1\text{-exp}(-z/D))$, where $T_{off,\infty}$ is the temperature offset at infinite depth, and $D$ scales its onset with depth.

Inspection indicates that $T_{off,\infty}$ can be estimated as a function of amplitude, wavelength, and difference between geotherm and

lapse rate as $T_{off,\infty}$ (est) = -2.65 $H_0^2$ ($g$-$\beta$)/$\lambda$ (Fig. 2C), and $D$ is a function of wavelength and amplitude, and can be estimated empirically as $D$(est) = 0.073 $\lambda$ + 0.22 $H_0$ (Fig. 2D). The former correction is imprecise at larger offsets, but even in the worst, severe case (8 km wavelength, 1.5 km amplitude) the correction is only off by 1.1 °C.

The depth correction for 1D solutions of temperature at the topographic mid-slope (subscript $m$) is thus:

$$T_m(z) = T_{1D}(z) - \frac{2.65\, H_0^2\, (g-\beta)}{\lambda}\left[1 - e^{-z/(0.073\lambda+0.22H_o)}\right] \tag{4}$$

The resulting correction is 0-10°C across conditions appropriate for modelling in HeFTy. Although fairly small in absolute

temperature terms, this correction also improves the calculation of the geothermal gradient at the mid-slope position, which turns out to be useful in the next section.





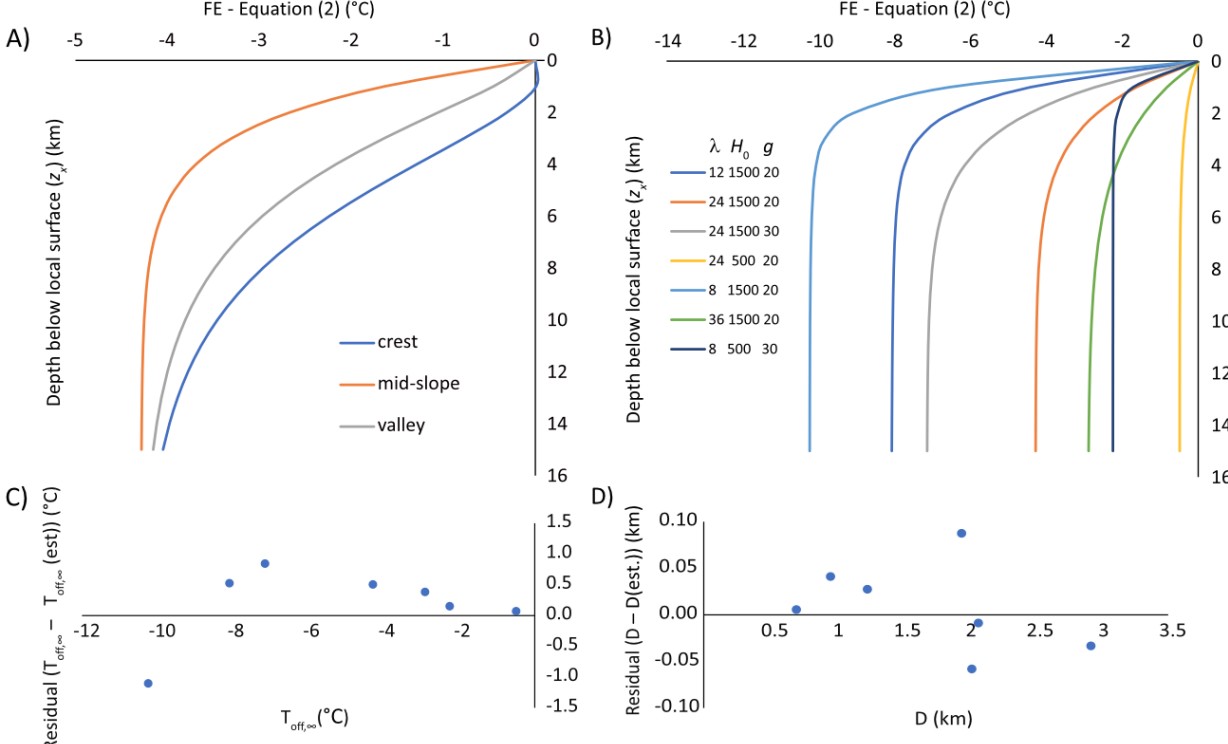

**Figure 2. A) Temperature difference in the steady state geotherm between finite element model and Equation (2) solution below the crest, mid-slope, and valley of topography with wavelength 24 km, amplitude 1500 m and heat flow corresponding to a steady-state geotherm of 20 °C/km. B) Mid-slope anomalies for models with varying wavelength, amplitude, and geotherm. C) Residuals for maximum temperature offset estimate. D) Residuals for depth scaling estimate.**

## 3.2 Steady state geotherm with erosion and constant topography

In general, achieving a steady state temperature regime from a standing start (i.e., going from no erosion to a constant rate) takes tens of millions of years, at least for erosion rates capable of creating and sustaining topography. It can be taken as a
given that a steady state is seldom achieved. But, once conditions are set up, it is useful as a marker for where isotherms are converging toward.

Stüwe et al. (1994) and Mancktelow and Grasemann (1997) derived analytical solutions for steady-state thermal structure beneath constant periodic topography and constant erosion. The Mancktelow and Grasemann (1997) formulation for erosion occurring at rate $u$, omitting radiogenic heating and keeping the $z_x$ shift, contains the same three elements as Equation (2), in
altered form:

$$T = T_0 + \gamma\left[1 - e^{-u\,z/\kappa}\right] + \left(\frac{u}{\kappa}\gamma - \beta\right) H_0 \cos\left(\frac{2\pi x}{\lambda}\right) e^{m_2 z_x} \tag{5}$$





Where $\kappa$ is thermal diffusivity, and for a constant basal temperature boundary condition at depth $L$ (as a constant basal gradient condition never converges to a steady state with continuous erosion):

$$\gamma = \frac{T_S - T_L}{1 - e^{-Lu/\kappa}} \tag{6}$$

where $T_S$ and $T_L$ are the constant temperature at the mean surface and at depth $L$, respectively. Also:

$$m_2 = -\frac{1}{2}\left[\frac{u}{\kappa} + \sqrt{\left(\frac{u}{\kappa}\right)^2 + \left(\frac{4\pi}{\lambda}\right)^2}\right] \tag{7}$$

The $m_2$ root term shows that the transition from surface to geothermal gradient occurs over a shallower interval than the no-erosion case, as for any positive $u$ it has a higher magnitude than the corresponding exponential term in Equation (2). At the same time, the magnitude of the isotherm displacement rises, as the steady state gradient given by $u\gamma/\kappa$ is higher than the no-erosion condition.

### 3.3 Geotherm evolution at crests and valleys with erosion and constant or growing topography

The different offsets from the analytical solution for the geotherm between the ridge and valley of topography in Figure 2A indicate that different corrections are necessary for each location. Figure 3 shows different views of the temperature difference at the same $z_x$ between the mid-slope versus the ridge (left; negative values) and valley (right; positive), at 1 m.y. intervals for a 2D finite element model with 24 km wavelength, 1500 m deflection, and 0.6 mm/yr erosion. The constant displacement between the curves at 0 km reflects the unchanging topography and lapse rate. The bulging of the curves with increasing time since onset of erosion reflects the bunching of isotherms toward the surface due to exhumation, which diminishes with depth but grows slowly through time as the effect of the bulging propagates downward.

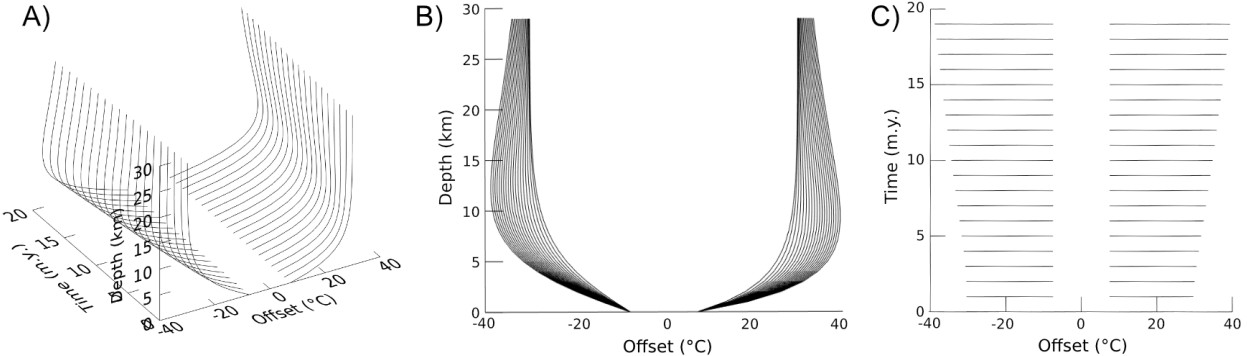

**Figure 3. Offsets of temperature with depth below local surface at periodic topography ridges (negative values) and valleys (positive values) compared to mid-slope (mean) elevation, at 1 m.y. intervals for model with parameters as in Figure 1b. A) Oblique view. B) Temperature offset versus depth. C) Temperature offset versus time.**





The diagrams illustrate that the change in temperature with depth and time due to topographic deflection has three components: maximum magnitude at great depth, magnitude variation with depth, and magnitude evolution with time. These are separable. The depth of the transition from the surface to maximum magnitude does not change through time and is dominantly controlled by the topographic wavelength, and the bulging is from the bunching of isotherms near the Earth surface as exhumation

progresses (Fig. 1b). The maximum magnitude at depth increases through time, and shows no sign of converging.

The progressive bunching of isotherms also occurs at the mid-slope position, suggesting that the mid-slope geotherm can be used as a normalizing factor. Dividing the offsets by the current local geotherm at the same z-t position beneath the mid-slope location ($dT/dz_m$ ($z$)) leads to Figure 4:

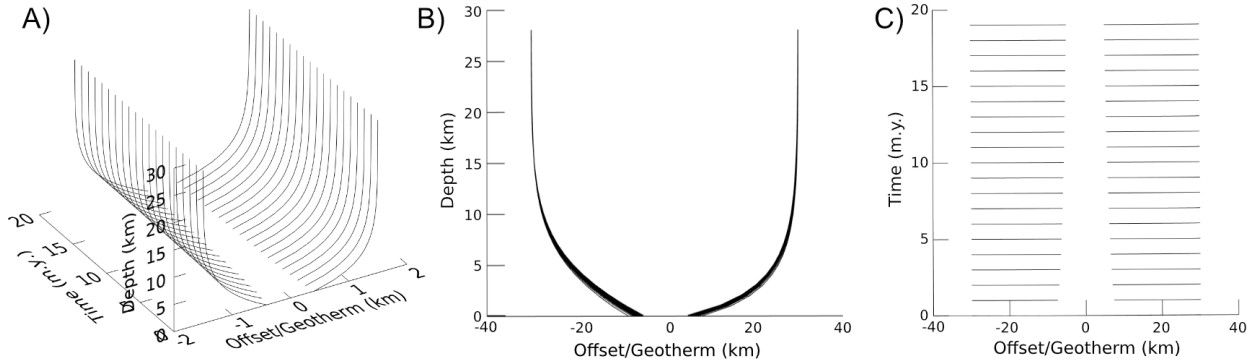

**Figure 4. Normalized offsets with depth below local surface at periodic topography ridges (negative values) and valleys (positive values) compared to mid-slope (mean) elevation, at 1 m.y. intervals for model with parameters as in Figure 1b. A) Oblique view. B) Normalized offset versus depth. C) Normalized offset versus time.**

It is evident that the time evolution of the offsets is almost entirely characterized by the time evolution of the geotherm at the mid-slope (mean elevation) point. There is a slight evolution in the nearest-surface points, but this is due to the evolution of the geotherm relative to the constant lapse rate. Figure 4 further clarifies the asymmetry between the ridge and valley offsets, indicating that they require separate treatment.

The same test can be run with growing topography, starting from no topography and evolving to amplitude 1500 m after 20

m.y. (Fig. 5). Normalization using the mid-slope geotherm again results in a regular pattern for the anomalies at the ridge and valley positions, with depth relationships closely resembling, and converging to, their counterparts in Figure 4. This result suggests that, regardless of the history of topographic development, the current state of topography and the current mid-slope geotherm adequately describe the time-varying component of the system.





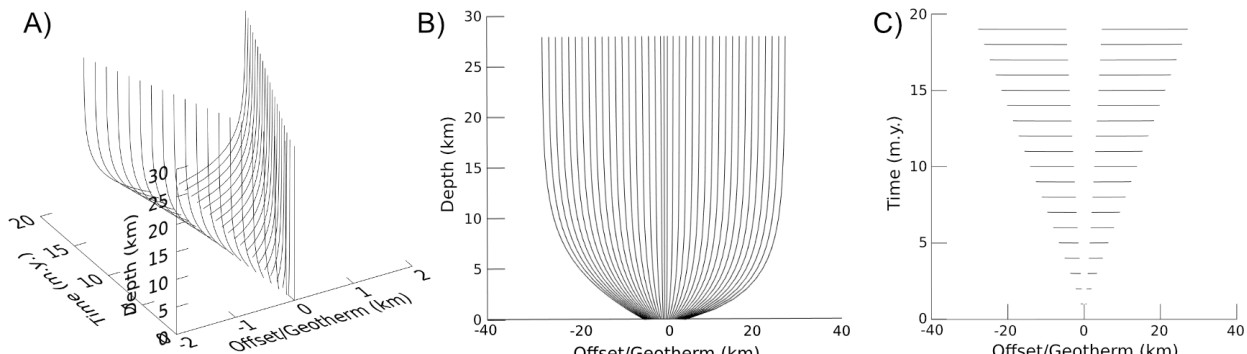

**Figure 5. As Fig. 4, for model with developing topography. A) Oblique view. B) Normalized offset versus depth. C) Normalized offset versus time.**

The asymmetric offsets in Figures 4B and 5B can be fitted well with an equation of the form:

$$T_{asym}(z) = \left(c_0 + c_1 e^{-\frac{z}{c_2}}\right)\left[\frac{dT}{dz}\right]_m (z) \tag{8}$$

in which $c_0$ defines the maximum normalized offset at depth, $c_1$ the normalized offset at the surface, and $c_2$ is the depth scale of the transition. It is readily apparent that $c_0$ corresponds to $H_0$, positive for valleys and negative for ridges, as it reflects the difference in temperature at depths where isotherms no longer deflect. Similarly, $c_1$ corresponds to the surface temperature offset, which is determined by $H_0$ and $\beta$. The set of relations for ridges (subscript $r$) and valleys (subscript $v$) are thus:

$$c_{0,v} = H_0$$

$$c_{0,r} = -H_0$$

$$c_{1,v} = -H_0\left(1 - \beta/\left[\frac{dT}{dz}\right]_m (0)\right) \tag{9}$$

$$c_{1,v} = H_0\left(1 - \beta/\left[\frac{dT}{dz}\right]_m (0)\right)$$

with $H_0$ representing current amplitude at a given time (i.e. during topographic development) in units of km, and geotherm and lapse rate in °C/km. Equations (8) and (9) combined provide:

$$T_{asym}(z) = \pm H_0\left\{1 - \left(1 - \beta/\left[\frac{dT}{dz}\right]_m (0)\right)e^{-\frac{z}{c_2}}\right\}\left[\frac{dT}{dz}\right]_m (z) \tag{10}$$

where the sign is positive for valleys and negative for ridges, and the $c_2$ variable changes between these cases.



To explore the depth scaling ($c_2$), a series of FETKin models varying in wavelength, amplitude, background steady-state
geotherm, erosion rate, and developing versus constant topography was used. Model settings are listed in Table 1, as well as
the corresponding $c_2$ values for ridges and valleys fitted to the model outcomes.

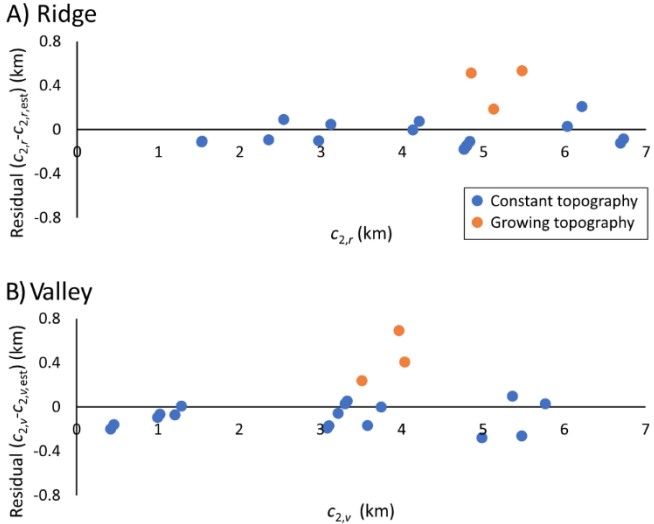

**Figure 6. Residual plots for estimates of depth scaling parameter
for ridges and valleys.**

Table 1. FETKin models for used to derive empirical relations between physical parameters and isotherm deflection during constant erosion

| Model # | 1 | 2 | 3 | 4 | 5 | 6 | 7 | 8 | 9 | 10 | 11 | 12 | 13 | 14 | 15 | 16 | 17 | 18 | 19 | 20 |
|---|---|---|---|---|---|---|---|---|---|---|---|---|---|---|---|---|---|---|---|---|
| $Q_m/K$ (°C/km) | 30 | 30 | 20 | 20 | 20 | 20 | 20 | 20 | 20 | 20 | 30 | 30 | 20 | 20 | 20 | 20 | 20 | 20 | 20 | 20 |
| $\lambda$ (km) | 24 | 24 | 24 | 24 | 36 | 36 | 12 | 12 | 8 | 8 | 8 | 8 | 36 | 36 | 24 | 24 | 24 | 24 | 24 | 24 |
| $H_0$ (km) | 1.5 | 1.5 | 0.5 | 0.5 | 0.5 | 0.5 | 1.5 | 1.5 | 1.5 | 1.5 | 0.5 | 0.5 | 1.5 | 1.5 | 1.5 | 1.5 | 1.5 | 1.5 | 1.5 | 0.75 |
| $u$ (mm/yr) | 0.0 | 0.6 | 0.0 | 0.6 | 0.0 | 0.6 | 0.0 | 0.6 | 0.0 | 0.6 | 0.0 | 0.6 | 0.0 | 0.6 | 0.0 | 0.6 | 0.3 | 0.3 | 0.6 | 0.3 |
| topography* | c | c | c | c | c | c | c | c | c | c | c | c | c | c | c | c | c | g | g | g |
| *depth scale from fitting model results (km)* | | | | | | | | | | | | | | | | | | | | |
| *ridge* $c_2$ | 4.80 | 4.76 | 4.13 | 4.21 | 6.03 | 6.21 | 3.12 | 2.97 | 2.54 | 2.36 | 1.54 | 1.53 | 6.69 | 6.72 | 4.83 | 4.77 | 4.79 | 5.13 | 5.47 | 4.85 |
| *valley* $c_2$ | 3.08 | 3.30 | 3.58 | 3.75 | 5.48 | 5.77 | 1.21 | 1.29 | 0.42 | 0.46 | 1.00 | 1.02 | 4.99 | 5.36 | 3.10 | 3.33 | 3.22 | 3.51 | 3.96 | 4.04 |
| *maximum absolute error in upper 10 km (°C)* | | | | | | | | | | | | | | | | | | | | |
| ridge | 0.8 | 1.6 | 0.1 | 0.3 | 0.1 | 0.3 | 1.2 | 1.7 | 1.5 | 2.1 | 0.3 | 0.6 | 0.3 | 0.9 | 0.5 | 1.0 | 0.7 | 1.0 | 1.6 | 0.6 |
| valley | 1.6 | 2.1 | 0.2 | 0.1 | 0.2 | 0.1 | 1.2 | 2.0 | 1.6 | 2.7 | 0.6 | 1.0 | 0.7 | 0.7 | 0.9 | 1.2 | 1.1 | 1.3 | 2.5 | 0.8 |
| *median absolute error in upper 10 km (°C)* | | | | | | | | | | | | | | | | | | | | |
| ridge | 0.6 | 1.2 | 0.0 | 0.1 | 0.0 | 0.2 | 0.6 | 1.5 | 0.7 | 1.5 | 0.2 | 0.3 | 0.2 | 0.6 | 0.4 | 0.7 | 0.6 | 0.6 | 1.0 | 0.5 |
| valley | 0.4 | 1.2 | 0.1 | 0.1 | 0.1 | 0.0 | 0.4 | 1.3 | 0.1 | 0.8 | 0.0 | 0.2 | 0.5 | 0.4 | 0.2 | 0.8 | 0.4 | 1.0 | 2.2 | 0.6 |

* c = constant topography, g = growing topography

The exponential parameter $c_2$ is dominantly a function of the topographic wavelength, and secondarily of topographic
amplitude. A good empirical fit is given by:





$$c_{2,v} = 0.166\,\lambda - 0.473\,H_0$$

$$c_{2,r} = 0.156\,\lambda - 0.805\,H_0 \tag{11}$$

As shown in Figure 6, these formulas do an excellent job of reproducing $c_2$ for a wide range of conditions. The principal apparently systematic departure is for growing versus static topography, with the former increasing the depth scaling by ~0.5 km compared to similar constant-topography models.

Although there is always a danger that empirical relations can degrade if applied outside of the model parameter space used to develop them, the scenarios used here reasonably bracket the use cases appropriate for modelling in HeFTy.

Table 1 also lists the maximum and median absolute temperature differences over the top 10 km of the crust between finite element models and estimates using the FE-calculated mid-slope geotherm and Equations 2 and 10. All errors at all times are less than 3 °C, with an average maximum error among models of 1 °C, and the average median error is 0.6 °C. The growing-topography scenarios also show small errors, suggesting that adjusting for such scenarios is a second-order consideration.

Further testing shows that these low errors are also achieved when exhumation rates vary. For example, a model akin to that 165 shown in Fig. 3 in which erosion proceeds for 10 m.y. at 0.6 mm/yr and then slows to 0.3 mm/yr over the subsequent 30 m.y. yields a maximum error of 1.2 °C and a median error of 0.8 °C, no different than the constant-rate model in Table 1 (model 16).

### 3.4 Interpolating across topography

Interpolating deflection between ridges and valleys is done via the cosine term in Equation (1). If ideal sinusoidal topography 170 is assumed then elevation $E$ varies around mean elevation $E_m$ as:

$$E = E_m + H_0 \cos\left(\frac{2\pi x}{\lambda}\right) \tag{12}$$

allowing the cosine term to be calculated as:

$$\frac{E - E_m}{H_0} = \cos\left(\frac{2\pi x}{\lambda}\right) \tag{13}$$

The complete equation for temperature asymmetry then becomes:

$$T_{asym}(z, x) = (E_m - E)\left\{1 - \left(1 - \beta\Big/\left[\frac{dT}{dz}\right]_m(0)\right)e^{-\frac{z}{c_2}}\right\}\left[\frac{dT}{dz}\right]_m(z) \tag{14}$$

### 4 Implementation in HeFTy

The foregoing calculations have been implemented into HeFTy (version 2.2.0 and higher) to provide a better basis for thermal 175 history modelling across elevation transects. In the new implementation, the position of the 1D thermal model has been changed from the "control" sample location, against which the positions of the other samples are determined in models





involving topographic development or tilting (Ketcham, 2024), to the mid-slope. This reduces the impact of choice of control sample. Whereas part of the need for carefully choosing the control sample stemmed from avoiding generating group exhumation paths that result in some samples being sent above the ground surface prior to the end of the model, the program now flags these cases during forward modelling, and avoids them automatically during inverse modelling.

### 4.1 Guidelines for data entry

As with other multi-sample modelling in HeFTy, it is best to run individual models on each sample prior to combining them, as an initial check for their respective viability and mutual consistency. Regional topographic parameters of wavelength, amplitude, and mean elevation are specified independently, and all samples should lie within the topographic range specified (mean elevation ± amplitude). The wavelength should correspond to the full wavelength of the region being modelled, even if the sample set does not include the peak and alley positions. Although position along the transect is entered via geographic coordinates, it is only used for display purposes, and HeFTy uses sample elevation to define its location along the topographic period for the thermal correction according to Equations (13) and (14).

### 4.2 Demonstration of topographic correction

Figure 7 shows a series of models results following from those used by Ketcham (2024, Fig. 8) without and then with the periodic topographic correction. In Fig. 7A, simple exhumation under constant (static) topography results in a series of parallel paths when there is no correction, as each path differs only in its surface boundary condition, defined by the lapse rate. Applying the topographic correction in Fig. 7B allows the specimens to transition from depth, where they are separated by ~25 °C, to the surface where their separation is 9 °C. Note that at the starting depth of 7 km the steady-state isotherms are not fully flat (see Fig. 1A), and thus the temperature offset among samples in the initial condition (steady state) does not quite correspond to the geothermal gradient. If topography begins flat at 40 Ma and develops during exhumation, the difference between the non-corrected and corrected path sets (Fig. 7C and D, respectively) is very subtle, with a maximum temperature difference of only about 2 °C. This demonstrates that topographic development has served as a reasonable proxy for periodic topography effects in previous studies using topographic profile modelling in HeFTy (Hestnes et al., 2024; Mackaman-Lofland et al., 2024), but that the models cannot distinguish whether topographic development was really required to fit the data.

Results are similar when tilting is included, in this case with a 10° clockwise rotation from past to present such that the topographic high corresponds with the uplifted side of the profile. Under constant topography the periodic correction again makes a difference of up to 10 °C, only this time the dispersion of paths decreases at depth because the topographic effect is inverted (Fig. 7E, F). If topography starts flat, then differences are more subtle, basically because there is no correction to make at the model start, and models all converge to the same ending point.

With regard to tilting, it should be noted that the 1D model does not incorporate the additional deflection of isotherms up or down due to block rotation; HeFTy only projects samples higher or lower in the crust. The error stemming from this omission



has been minimized by making rotational pivot the mid-slope rather than the control sample, and is expected to usually be second-order.

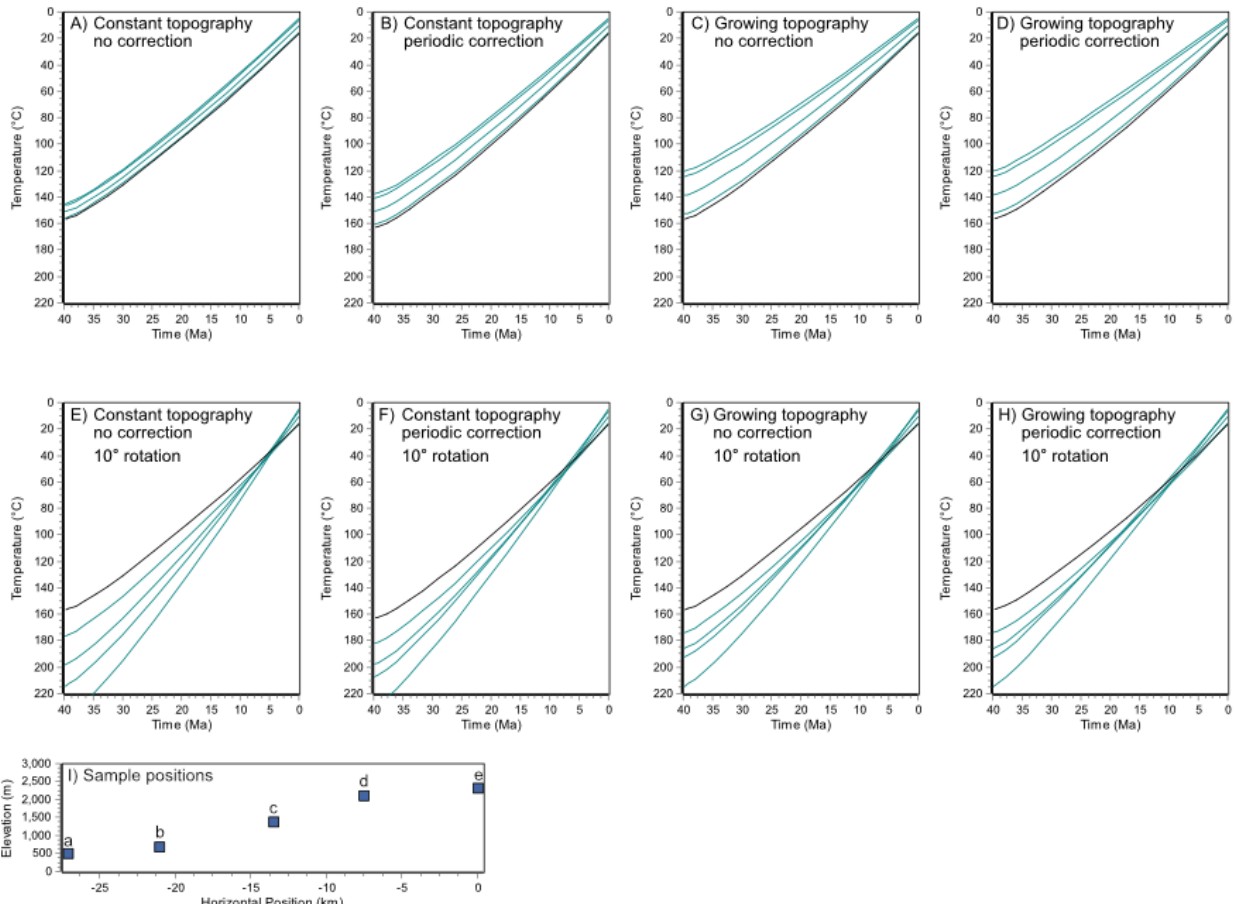

**Figure 7. Effects of topographic correction.** Each model shows the set of t-T histories assuming the control sample (sample a in all cases) is exhumed from 7 km depth to the surface over a 40 m.y. span. For all HeFTy models, MSL temperature is 20 °C, basal heat flux is 60 mW m$^{-2}$, thermal conductivity 3 W m$^{-1}$ K$^{-1}$, heat capacity 800 J kg$^{-1}$ K$^{-1}$, density 2700 kg m$^{-3}$, maximum depth 120 km, grid spacing 2 km, and β 6.5 °C km$^{-1}$. For the periodic correction, λ=54 m, $H_0$ = 915 m, and $E_m$ = 1415 m. In all cases, the black curve is for sample a, and the teal curves are for the others, with sample e always the most distant. Models shown in A, C, E, and G do not use the periodic topography correction, which can be compared with models B, D, F, and H which do.

## 5 Discussion

As shown in the example, the absolute temperature changes enabled by the periodic correction developed here will tend to be modest, in many cases less than 10 °C, although the absolute value will depend on the severity of topography. The primary



utility in this correction, aside from increasing the overall accuracy of the solution, is to better allow topographic effects to be "felt" at depth during exhumation, in turn increasing the ability for the data to discern when topographic evolution began. In

the context of multi-sample thermal history modelling, even relatively mild relative excursions among samples that are geographically linked can significantly affect their ability to have their thermochronometric data be matched in tandem.

It should also be stressed that this functionality is meant to test relatively simple scenarios and provide a first-order approximation of the geological processes being considered; it should not be seen as depicting or reproducing reality. In particular, in studies where the exhumation history is more complex, such as involving spatially variable and/or significant 3D

topographic development, or directly impacted by additional processes such as folding or heat transfer across faults or due to fluids, more sophisticated modelling tools will be preferable for posing questions with the appropriate level of fidelity and detail (e.g., Almendral et al., 2015; Braun, 2003; Bernard et al., 2022).

## Author Contributions

RAK executed all aspects of this study.

## 225 Code Availability

HeFTy is written in a commercial programming environment. The compiled program is available free of charge to academic, not-for-profit users.

## Competing Interests

The author declares that he has no conflict of interest.

## 230 Acknowledgements

Preparation of this paper was supported in part by the Geology Foundation of the Jackson School of Geosciences.

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
