# Peer review of "Technical Note: Incorporating topographic deflection effects into thermal history modelling"

_EGUsphere, 2025_

## Referee Comment (RC2)

I went back to Mancktelow and Grasemann (1997) where it is stated that no solution exists to the heat conduction-advection equation if one assumes that the temperature increases linearly with depth when depth goes to infinity. I agree with this. However, the suitable solution that we seek here is that corresponding to a given heat flux coming from the mantle, i.e., at some depth (the thickness of the lithosphere?). So this correspond to imposing a temperature gradient at a given depth (without making any assumption about what happens below). In the Earth the base of the lithosphere may be considered as a boundary between two domains, one where conduction dominates over advection and the other where advection dominates over conduction.

The point I made in my earlier review of this (very good) manuscript is that there exists an analytical solution to the steady-state conduction-advection heat equation with a basal heat flux (or gradient) condition. The equation that I provided was quite cryptic. Below I give it in a clearer form and demonstrate how it can be derived. I do so because I think this is an important (and general) point to make for our quantitative interpretation of thermchron data.

The conductive-advective heat transport equation at steady-state represents a balance between two terms, one representing advection of heat by rocks travelling towards the surface and the other representing their conductive cooling:

$$-v\frac{\partial T}{\partial z} = \kappa\frac{\partial^2 T}{\partial z^2} \tag{1}$$

where $T$ is temperature, $z$ is depth, $v$ is rock advection velocity towards the surface and $\kappa$ heat diffusivity.

Assuming the following boundary conditions:

$$T(0) = 0 \text{ and } \frac{\partial T}{\partial z}(L) = G \tag{2}$$

we can express the same equation in dimensionless form:

$$-P_e\frac{\partial T'}{\partial z'} = \frac{\partial^2 T'}{\partial z'^2} \tag{3}$$

$$T(0) = 0 \text{ and } \frac{\partial T'}{\partial z'}(1) = 1 \tag{4}$$

using the following variables:

$$z' = z/L, \ T' = T/GL \text{ and } P_e = \frac{vL}{\kappa} \tag{5}$$

The general solution to this second-order equation is:

$$T(z') = C_1 e^{-P_e z'} + C_2 \tag{6}$$

Adding the surface boundary condition:

$$T(0) = 0 \rightarrow C_1 + C_2 = 0 \tag{7}$$

yields the following form:

$$T(z') = C_1(1 - e^{-P_e z'}) \tag{8}$$

Adding the second boundary condition:

$$\frac{\partial T'}{\partial z'}(z' = 1) = 1 \rightarrow C_1 = \frac{1}{P_e e^{-P_e}} \tag{9}$$

we obtain the steady-state solution in dimensionless form:

$$T(z') = \frac{1}{P_e} \frac{1 - e^{-P_e z'}}{e^{-P_e L}} \tag{10}$$

or in its original (dimensional) form:

$$T(z) = \frac{G\kappa}{v}\left(\frac{1 - e^{-vz/\kappa}}{e^{-vL/\kappa}}\right) \tag{11}$$

which satisfies the initial equation and the two boundary conditions.

In the following figure, I show a numerical solution to the transient equation:

$$\frac{\partial T}{\partial t} - v\frac{\partial T}{\partial z} = \kappa\frac{\partial^2 T}{\partial z^2} \tag{12}$$

assuming an initial condition at conductive steady-state:

$$T(z) = \frac{z\, T_l}{L} \tag{13}$$

where $T_l = 500°\mathrm{C}$, $L = 30$ km, $\kappa = 25$ km/Myr$^2$, $v = 1$ km/Myr and $G = T_l/L$. I used an implicit finite difference scheme to solve this equation. I show ten steps during the transient stage, as well as the initial (conductive) solution and the analytical steady-state solution derived above for comparison. We see that the numerical solution does indeed converge towards the analytical solution as time increases.

[Figure]

Figure 1: Comparison between numerical transient solution (colored lines) and analytical steady-state solution (black line with crosses).

---

## Author Comment (AC1)

Response to the review of J. Braun.  Original comment in black, responses in red.
Richard Ketcham, 19 June 2025.

I have read the technical note entitled: "Incorporating topographic deflection effects into thermal history modelling" by Richard Ketcham with great interest. I believe this is a very useful contribution to the field of quantitative thermochronology, especially because it is incorporated in a widely used software for the interpretation of thermochron data (HeFTy).
I thank the reviewer for the endorsement.

 The note proposes semi-analytical solutions, i.e., combining approximate analytical solutions to the 2D heat equation to empirical relationships derived, for the most, from numerical (finite element) solutions of the same equation. These solutions provide useful corrections for the cooling history of rocks being exhumed to the Earth's surface that are then used to compute thermochronological ages and other quantities to be compared with observational constraints.

 I have a few points that I believe need to be addressed (all of a technical nature) and that would greatly help future users of the software in their understanding of the method and the advantages and limitations of its use in the interpretation of their data.

1. The corrections that are proposed here are meant to represent the effect of a finite amplitude, potentially time-dependent, topography on the geometry of the underlying isotherms, including the effect of vertical advection. As demonstrated by the authors these effects can be substantial. However, little mention is made of the effect of horizontal advection, which can be, in many cases, dominant over other effects. Indeed, motion of rock particles parallel to dipping faults can cause them to experience cooling paths that are drastically different from those obtained assuming pure vertical advection. Example of this can be seen in our interpretation of thermochron data in the Southern Alps of New Zealand (I refer here to the PhD works of G Batt and F Herman under my supervision many years ago). I do not suggest here that this perturbation be added to the current note/work, but that this fact should be made clear to the user.
I had intended for this point to be covered in the final sentence of the discussion, but had not directly mentioned horizontal advection, so I've adjusted the text to do so in the revised version.

2. The proposed correction(s) are all based on the assumption that the topography has a single wavelength. However, surface topography is often composed of more than a single wavelength. It would be very useful to better describe what the user should use as a topographic wavelength and amplitude in a given situation. For this, it would be useful to indicate that each terhmochronometric system (as defined by its closure temperature or temperature sensitivity) is sensitive to different topographic wavelengths because the depth (and thus the temperature) to which the topographic perturbation propagates is strongly (exponentiallY) dependent on the wavelength. This warrants a short paragraph in this report and, potentially in the user interface of the software, to help the user decide what wavelength and which amplitude is to be used in a real case application.
This is an excellent point.  A short paragraph has been added to section 4.1 on data entry in HeFTy.

3. Alternatively, the software could be adapted to use a multi-wavelength topography but this would require that the corrections (for each wavelength and amplitude) be simply combined (added?). This would require additional work to implement and could be quite useful, but potentially beyond the scope of this short note. I leave it to the author to appreciate whether this should be done.

This is an interesting suggestion, but it is indeed beyond the scope of this note. It's not clear if there is a straightforward solution using the approach developed here; I imagine one would have to enter not only the wavelengths and amplitudes but also offsets of the different components.

4.My last point, and maybe the most critical one, concerns the time-dependent solution. If I understand well the author proposes to use the steady-state solutions derived analytically and empirically to cases where the topography grows with time by simply using a time varying value for $H_0$. As shown by the author, this seems to provide relatively good results but my suspicion is that this is because the rate of topographic change remains relatively low compared to the rate of heat diffusion, i.e., the cases he explores must be at relatively low Peclet number. I suspect that a very rapid incision event (that would make the topographic relief grow rapidly with time) would produce a thermal response that cannot be adequately approximated by a series of steady-state solutions. Again, I do not suggest to the author to improve his solution but to warn the users of the limitation (in terms of how fast topography can change) of the method proposed here.

This study did not use steady state solutions, except to set up the initial conditions. The 1D model is used to characterize main part of the time-varying component of the system. Evidently this was not brought out clearly enough, so I've moved some text from section 3.3 to 3.4, added a bit more, and expanded the title of 3.4, to emphasize that this is a dynamic solution.
I also added a test case for very rapid incision (3 km of relief with an 8 km wavelength in 2 million years) and documented the larger discrepancies that arise (which are still modest if considered in terms of vertical or temporal displacement). New text places this test in the context of advising care and considering using other approaches when working beyond the parameter space used to generate this model.
In the course of making these changes, I also found a typo in equation (11), where a minus should have been a plus; this is fixed in the revised text, but the underlying code was/is correct.

 I also have a few minor comments:

a. I have tried to reproduce the predictions of this semi-analytical approach to compare them to solutions of the heat equation obtained by Pecube but failed to do so because the value of some constants, or the type or value of basal boundary conditions were not clearly given. It would be good if the author could include all the information necessary to reproduce the results shown in the note.

The basal flux is added in the revised text.

b. I do not think there is an analytical solution of the heat equation in Fox et al, 2014.

The Fox citation without an analytical solution has been removed (they were actually both 2014 papers, and one was erroneously cited as being from 2013).

c. It is not clear what the "relative vertical rotation" mentioned at line 38 is; a few words of explanation would be useful (or a small sketch).

Citation added for where diagram and explanation provided.

d. On line 78, there is a reference to "Finite element modelling"; this sounds like it requires a reference or it should be modified to "Our finite element modelling, shown in Figure X", if this is indeed the case.

The revised text clarifies that this was the finite element modeling performed from this study. The temperature offset is illustrated in Figure 2.

e. On Line 106 the statement that "a constant basal gradient condition never converges to a steady-state with continuous erosion" is not correct, in my opinion. In 1D, the solution is:

$$T(z) = T_0+\frac{G\kappa}{u}e^{Lu/\kappa}(1-e^{zu/\kappa})$$

where $G$ is the assumed basal gradient at depth $L$.

The equation provided by the reviewer may be incomplete, as when I attempt to plug in some numbers it does not produce the desired result; in particular, its derivative (G e^(L z u/kappa)) does not produce a gradient of G at depth L unless the erosion rate u is negligible. However, assuming that there's another solution, this may be a question of how one defines and enforces the condition. Mancktelow and Grasemann (1997) claim that a steady state solution is only possible if the temperature approaches a constant value at depth, and my statement was essentially agreeing with this; with a constant-flux boundary condition and no other enforcement, the basal temperature rises continuously. But, if one sets a constant temperature at depth, eventually a stable steady state gradient will arise associated with it. To resolve the ambiguity, I've replaced my statement with Mancktelow and Grasemann's.

---

## Author Comment (AC3)

Response to the review of C. Glotzbach.  Original comment is in black, responses in red.
Richard Ketcham, 8 July 2025

I enjoyed reading your manuscript. It presents an important contribution to the community, and given that this functionality is implemented in **HeFTy**, I am confident it will be widely used in the future to enable more accurate interpretation of thermochronological data.

The technical note is well written and pitched at a level that will be understandable and useful to the intended audience from the thermochronological community.

I thank the reviewer for the endorsement.

One point I would encourage the authors to consider relates to the **automatic assignment of samples to positions along the sine-shaped topography**. While the approach is elegant, real topography is often more complex. I could foresee potential issues where samples collected from short-wavelength features (e.g., secondary peaks) might be incorrectly positioned, particularly in landscapes where the dominant wavelength is much larger. A brief discussion of this limitation or potential strategies to mitigate it would strengthen the manuscript.

Agreed.  In the revised version I've added another sentence to the data entry section discussing this limitation.

Please also see my **technical comments:**

**Technical corrections:**

Line 47-50: A quite similar dataset was compiled by Glotzbach et al. (2015) in combination with a Fourier approach to empirically estimate the perturbation of isotherms in complex 2D-3D topographic situations. You may want to cite this work.

This is a good reference to add, but it goes better with the new text added in response to the other reviewer's comments about multiple wavelengths and their potential effects on lower-Tc systems.

Line 54-55: The choice of parameterisation would require some justification or showing a few alternative models, e.g. running until steady-state or with a different thermal conductivity. This would allow for the estimation of the uncertainty introduced by the choice of parameterisation.

One can always run more models.  I'm not sure what running until steady state would contribute; as demonstrated by Fig. 4, the mid-slope-normalized offset barely evolves in time.  The mid-slope normalization also accounts for differences in thermal parameters.  Changes in conductivity and diffusivity are effectively equivalent to changes in geotherm and exhumation rate, respectively (via Eq. 1, 5, 6, 7), and so the large variations modeled in the latter should adequately cover the comparatively modest expected variations in the former.  A sentence expressing the latter point has been added to section 3.3.  I also note that Glotzbach et al. (2015) did not mention varying thermal parameters in their 142,000 models.

Line 155: This is not very clear, maybe you can also give the relative deviation and have a figure showing the difference between numerical and empirical solution, taking into account the uncertainty in c2.

Table 1 already provides the difference between numerical and empirical solutions, the latter based on both c2 and all of the other equations.

Line 211-212: Would be good to investigate under which boundary conditions the temperature difference is larger than 10°C and report this, to prevent users from stating that they can model their data without taking into account the topographic deflection effects

I've clarified that 10°C really means ±10°C, or a range of 20°C, which hopefully will grab more attention.